# Virus Prevalence in Egg Samples Collected from Naturally Selected and Traditionally Managed Honey Bee Colonies across Europe

**DOI:** 10.3390/v14112442

**Published:** 2022-11-03

**Authors:** David Claeys Bouuaert, Lina De Smet, Marleen Brunain, Bjørn Dahle, Tjeerd Blacquière, Anne Dalmon, Daniel Dezmirean, Dylan Elen, Janja Filipi, Alexandru Giurgiu, Aleš Gregorc, John Kefuss, Barbara Locke, Joachim R. de Miranda, Melissa Oddie, Delphine Panziera, Melanie Parejo, Maria Alice Pinto, Dirk C. de Graaf

**Affiliations:** 1Department of Biochemistry and Microbiology, Ghent University, 9000 Ghent, Belgium; 2Faculty of Environmental Sciences and Natural Resource Management, Norwegian University of Life Sciences, 1430 Ås, Norway; 3Norwegian Beekeepers Association, 2040 Kløfta, Norway; 4Bees@wur, Wageningen University & Research, 6708 Wageningen, The Netherlands; 5Abeilles et Environnement, INRAE, 84914 Avignon, France; 6Department of Apiculture and Sericulture, University of Agricultural Sciences and Veterinary Medicine, 400372 Cluj-Napoca, Romania; 7Department of Molecular Ecology & Evolution, School of Natural Sciences, Bangor University, Bangor LL57 2DG, UK; 8Taskforce Research, ZwarteBij.org VZW, 9890 Gavere, Belgium; 9Department of Ecology, Agronomy and Aquaculture, University of Zadar, 23000 Zadar, Croatia; 10Faculty of Agriculture and Life Sciences, University of Maribor, 2311 Pivola, Slovenia; 11Le Rucher D’Oc, 31200 Toulouse, France; 12Department of Ecology, Swedish University of Agricultural Sciences, 75007 Uppsala, Sweden; 13Applied Genomics and Bioinformatics, University of the Basque Country, 48940 Leioa, Spain; 14Centro de Investigação de Montanha, Instituto Politécnico de Bragança, Campus de Santa Apolónia, 5300 Bragança, Portugal; 15Laboratório Associado Para a Sustentabilidade e Tecnologia em Regiões de Montanha (SusTEC), Instituto Politécnico de Bragança, Campus de Santa Apolónia, 5300 Bragança, Portugal

**Keywords:** honey bee, suppressed in ovo virus infection, vertical transmission, virus resistance

## Abstract

Monitoring virus infections can be an important selection tool in honey bee breeding. A recent study pointed towards an association between the virus-free status of eggs and an increased virus resistance to deformed wing virus (DWV) at the colony level. In this study, eggs from both naturally surviving and traditionally managed colonies from across Europe were screened for the prevalence of different viruses. Screenings were performed using the phenotyping protocol of the ‘suppressed in ovo virus infection’ trait but with qPCR instead of end-point PCR and a primer set that covers all DWV genotypes. Of the 213 screened samples, 109 were infected with DWV, 54 were infected with black queen cell virus (BQCV), 3 were infected with the sacbrood virus, and 2 were infected with the acute bee paralyses virus. It was demonstrated that incidences of the vertical transmission of DWV were more frequent in naturally surviving than in traditionally managed colonies, although the virus loads in the eggs remained the same. When comparing virus infections with queen age, older queens showed significantly lower infection loads of DWV in both traditionally managed and naturally surviving colonies, as well as reduced DWV infection frequencies in traditionally managed colonies. We determined that the detection frequencies of DWV and BQCV in honey bee eggs were lower in samples obtained in the spring than in those collected in the summer, indicating that vertical transmission may be lower in spring. Together, these patterns in vertical transmission show that honey bee queens have the potential to reduce the degree of vertical transmission over time.

## 1. Introduction

Disease pressure is an inherent driver of the evolution of eusociality [1,2] or social task division [3], and it forms an important component in the evolution of western honey bees (*Apis mellifera*). With the arrival of the Varroa mite (*Varroa destructor*), the virus landscape in honey bee colonies was considerably changed by the introduction of a new transmission pathway, thereby influencing virus virulence and evolution [4,5,6,7,8,9,10,11,12,13]. Through Varroa-mediated transmission, virus diseases have become one of the most important proximate causes of colony mortality and honey bee decline [14,15,16,17,18,19,20,21].

Of the 72 virus species that have been identified in honey bees [22], the most commonly occurring belong to the families Iflaviridae and Dicistroviridae [23], particularly the sacbrood virus (SBV), black queen cell virus (BQCV), acute bee paralysis virus (ABPV) and deformed wing virus (DWV). Both ABPV and DWV consist of a complex of closely related, co-circulating master variants capable of forming viable recombinants [22,24,25]. The DWV complex is best described as a group of functionally and genetically compatible minor and major variants and their recombinants based on four master strains [26], of which DWV-A and DWV-B are currently the most common [27,28,29,30,31,32]. Dynamics in the presence and abundance of honey bee viruses show strong seasonal and geographical variation [33,34,35]. This variation is driven by the local adaptations of the virus, host and vector species, as well as by the specific characteristics of each virus [36,37,38,39,40]. Together, they form a geographic mosaic of coevolution [41].

In the first years after managed colonies are left untreated against the Varroa mite, colony mortality increases considerably [42,43]. This results in strong selective pressure forcing bees, mites and the viruses to adapt to each other. Most naturally surviving populations consist of unmanaged or feral colonies [40]. In managed colonies, two approaches have been described to transition from treated colonies to naturally surviving colonies. The first consists of leaving a large number of colonies unmanaged with respect to swarming, re-queening and Varroa control [39] and allowing natural selection to take place. This is described as the ‘Bond’ test: ‘live and let die’ [39]. A second approach, named ‘Darwinian black box’, builds further on this by adding selection for strong spring development [43]. One of the best studied naturally surviving populations with regard to virus–host coevolution is an isolated, closed honey bee population located at the tip of the Näsudden peninsula in the south of Gotland, a Swedish island in the Baltic sea. After implementing the Bond test in 1999, these honey bee colonies evolved an increased tolerance for DWV infections [38,44]. In addition, BQCV and SBV infections were less abundant in the autumn and early spring, possibly due to the reduced colony size of the Gotland colonies in these seasons [45].

Each response to a parasite influences transmission dynamics within and between honey bee colonies. As honey bees live in large groups, the transmission of viruses through trophallaxis, feeding or body contact occurs frequently. This form of transmission between individuals of the same generation is defined as horizontal transmission. However, transmission between generations by either eggs or semen is defined as vertical transmission [27]. Virus infections of queens, or their eggs, have been shown to interfere with normal egg development, to elicit a stress response in eggs [46] and to cause important health risks for the queen herself [47,48,49,50]. The importance of the honey bee queen in the viral dynamics of the colony was recently highlighted with the discovery of the ‘suppressed in ovo virus infection’ trait (SOV) [49]. This trait is described by the virus status of a sample containing 10 pooled drone eggs, and it reflects the degree of vertical transmission of viruses at the time of sampling. Colonies headed by a queen laying virus-free eggs have been found to show fewer and less severe DWV infections in almost all developmental stages of both drones and workers [49]. In addition, this potential to suppress viral infections is heritable [49] and alters the tissue specificity of DWV [50]. Drone eggs are unfertilized and therefore only reflect the vertical virus transmission of the queen to her offspring [51]. As drone eggs are only produced in the spring and summer, the choice of drone eggs compared to worker eggs includes the limitation of when samples can be collected.

The aim of this study was to compare virus infections in eggs collected from naturally surviving colonies (NSCs) and traditionally managed colonies (TMCs) across Europe, along with an analysis of the effects that queen age and sampling season have on infection patterns in these eggs. The previously described SOV phenotyping protocol [49] was used to screen for the presence of viruses after implementing two improvements: First, the detection of viral pathogens using qPCR instead of *end-point* PCR allowed for the quantification of the viral load of the eggs and lowered the detection threshold. Second, as the SOV trait is associated with increased virus resistance across DWV genotypes [50], a shift was made from screening for DWV-A only to a generic screening for the DWV complex. Overall, this research improved our understanding of how the patterns of the vertical transmission of viruses differ across Europe and in different breeding program selection strategies.

## 2. Materials and Methods

### 2.1. Virus Screening in Eggs

The 187 samples collected as part of the Flemish bee-breeding program in 2020 were used to compare the transition from an *end-point* PCR to a qPCR approach and to compare the quantification of DWV infections using primers specific to DWV-A or DWV-B and a generic DWV primer (DWV-Fam). All samples were collected following the phenotyping protocol for the SOV trait, as described by de Graaf et al. (2020), and they were screened using qPCR for DWV-A, DWV-B, DWV-Fam, SBV, ABPV and BQCV. To compare the performance of the *end-point* PCR to that of the qPCR, positive samples covering a 10^1^–10^8^/10 egg range were selected and analyzed using *end-point* PCR for SBV, ABPV, DWV-A and BQCV.

### 2.2. Egg Sample Collection across Europe

Egg samples were collected in 9 countries across Europe during either the spring (the beginning of March to the end of May) or summer (the beginning of June to the end of July) of 2020, depending on the presence of drone brood in each country and in each season (Table 1). Each country sampled between 6 and 12 TMCs (colonies managed following local standard practices, including treatment against the Varroa mite) and, if present, between 4 and 13 NSCs (colonies from populations that survive without treatment against the Varroa mite). In total, 53 samples were collected from NSCs, and 160 were collected from TMCs (including 72 from Slovenia). The samples from Slovenia were collected in the scope of a different project, hence the larger sample size. The colonies were managed following local standard practices, and the queens descended from locally adapted or native stock. The treatment of the TMCs against the Varroa mite was performed with registered products in each country. The management of the NSCs was conducted according to the ‘Darwinian black box’ selection method [43] or the ‘Bond’ test [39]. From each of the 213 sampled colonies, a pooled sample of 10 drone eggs was collected following the phenotyping protocol of the SOV trait, as described by de Graaf et al. (2020). If drone eggs were not present in the spring or summer and if attempts to induce drone laying did not succeed, worker eggs were collected instead (as was the case for 34 colonies). All samples were immediately stored at −20 °C and kept in a cold chain during transport to Belgium, where they were analyzed for DWV-Fam, BQCV, SBV and ABPV using RT-qPCR. For each sampled colony, information was gathered on the sampling season (spring or summer), subspecies, queen age, beekeeping method (for both TMCs and NSCs) and the presence of clinical signs at the time of sampling. This information was used to explain possible outliers and to look for correlations between multiple factors. Additional samples were collected if the apiary was composed of more than 10 colonies with drone eggs present during sampling. Appendix A provides an overview of the sampled populations and countries, the number of worker egg samples collected in each country and the presence of clinical signs at the time of sampling, and it lists the location of the sampled populations, including the year of establishment of the NSC populations.

### 2.3. RNA Extraction and cDNA Synthesis

All samples were first homogenized in the presence of zirconium beads in 0.5 mL QIAzol lysis reagent (Qiagen, Hilden, Germany). RNA was extracted using an RNeasy Lipid Tissue Mini Kit (Qiagen) according to the manufacturer’s instructions, including a DNAse step, and it was finally eluted in 30 µL elution buffer. The concentration of the total RNA was measured using Nanodrop (Isogen, De Meern, The Netherlands). Using random hexamer primers, 200 ng of RNA was retro-transcribed with a RevertAid H Minus First Strand cDNA Synthesis Kit (Thermo Scientific, Waltham, MA, USA). Honey bee β-actin was used to control RNA integrity.

### 2.4. End-Point PCR

All *end-point* PCR reaction mixtures contained 2 μM of each primer (see Appendix A), 1 mM MgCl_2_, 0.2 mM dNTPs each, 1.2 U HotStarTaq Plus DNA polymerase (Qiagen) and 2 μL cDNA product. The *end-point* PCR assays were performed using the following cycling conditions: 95 °C for 5 min; 94 °C for 30 s, 55 °C for 30 s, 72 °C for 1 min, 35 cycles; final elongation 72 °C for 10 min, hold 4 °C. The *end-point* PCR amplicons were analyzed by electrophoresis using 1.5% agarose gels stained with ethidium bromide and visualized under UV light. Positive and negative controls were included in each run.

### 2.5. qPCR

The virus load qPCR determination was performed using Platinum™ SYBR™ Green qPCR SuperMix-UDG (Thermo Scientific). Each reaction consisted of 0.4 µM of each primer (sequences provided in Appendix A), 11.45 µL RNase-free water, 12.5 µL SYBR Green and 1 µL of cDNA template. All samples were run in duplicate in a three-step RT-qPCR. The thermal cycling conditions started with an initial activation stage at 95 °C for 2 min, followed by 35 cycles of a denaturation stage at 95 °C for 15 s, an annealing stage at 58 °C for 20 s and an extension stage at 72 °C for 30 s. This procedure was followed by a melting curve analysis to confirm the specificity of the product (55–95  °C with increments of 0.5 °C s^−1^). Each plate included a no-template control and a positive control. A standard curve obtained through an 8-fold 5× serial dilution of a known amount of viral plasmid loads (range of 10^4^–10^10^ copies/µL) was used for absolute quantification. All data were analyzed using CFX Manager™ 3.1 software (Bio-Rad). Baseline correction and threshold setting were performed using the automatic calculation offered by the same software. The maximum accepted quantification cycle (Ct) difference between replicates was set to two Ct. The successful amplification of the β-actin internal reference gene was used to confirm RNA integrity throughout the entire procedure [52]. For each sample, the virus load for the 200 ng RNA included in the cDNA reaction was multiplied to account for the total volume of RNA per sample, and it was subsequently divided by 10 to represent data as total viral load per individual egg. The linear standard equations for the plasmid standards and primers specific for each virus were as follows: Ct = −3.519 × +47.762, R^2^ = 0.997 for DWV; Ct = −4.458 × +50.946, R^2^ = 0.865 for ABPV; Ct = −4.260 × +55.151, R^2^ = 0.943 for BQCV; and Ct = −4.571 × +45.651, R^2^ = 0.938 for SBV.

### 2.6. Statistics

The viral loads for each sample were Log10-transformed to improve data visualization. Detection thresholds for all pathogens were set at 30 Ct (corresponding to 103 copies for DWV, BQCV, SBV and APBV). Below this threshold, samples cannot be reliably quantified using qPCR [53]. RStudio version 3.6.1 was used for data analyses and visualization. Analyses of the differences in the number of infections between groups were conducted using chi-squared tests. For comparisons between the infection loads, T-tests were used. All tests were checked for and complied with the required assumptions.

## 3. Results

### 3.1. SOV Phenotyping Method

The virus detection thresholds on the end-point PCR (based on the end-point PCR of the samples positive on the qPCR along a 10^1^–10^8^ copy/reaction range of the starting template) were 10^2^ for DWV-A, 10^7^ for BQCV, 10^6^ for ABPV and 10^8^ for SBV (see Appendix A). A total of 187 samples were screened for DWV-A, DWV-B and DWV-Fam (generic DWV primer). Of these, 153 (82%) showed amplification when screening with DWV-Fam. One sample was only amplified with the DWV-A assay, and two samples were only amplified with the DWV-B assay. Of the 153 samples that were amplified with the DWV-Fam assay, 98 (64%) were also amplified with the DWV-B assay, 2 (1%) were amplified with the DWV-A assay, and 6 (4%) were amplified with both the DWV-A and DWV-B assays. The remaining 47 samples (25%) were only amplified with the DWV-Fam assay and not with either the DWV-A or DWV-B assay. The median infection load for DWV-Fam was 5.8 Log_10_ virus copy number/egg and was on average 1.69 Log_10_ higher than the sum of DWV-A and DWV-B.

### 3.2. Virus Prevalence

Table 1 shows an overview of the different virus prevalence across all samples, and it presents the number of collected samples per country, the number of infections for each virus, and the mean infection load for each country and for each selection strategy (TMCs or NSCs). In total, 53 samples were collected from NSCs, and 160 were collected from TMCs (including 72 from Slovenia). Of the 213 pooled egg samples screened, most infections were with DWV (51%), followed by BQCV (25%). Only three samples were infected with SBV, and two samples were infected with ABPV. Multiple virus infections in the same sample occurred in only 14% of the samples (28 samples were infected with two viruses, and one was infected with three viruses). No virus infections were found in 35% (74/213) of the samples. The differences in infection frequencies varied considerably between and within countries. The worker egg samples from Norway and Sweden had lower infection frequencies (11/34) than the drone egg samples from other locations (61/107). The Slovenian samples were collected in the scope of a different project, hence the larger number of samples. To avoid an uneven distribution of the sample size across groups, the Slovenian samples were not included for further analyses in this study. The differences between subspecies could not be analyzed due to the high variability between countries and the hybridization between subspecies. Due to bad weather conditions or the inaccessibility of some locations, some countries (The Netherlands, Romania and Sweden) were not able to sample the requested number of colonies.

Figure 1 shows the percentage of virus infections occurring in the two sampling seasons (spring and summer). The samples collected in the spring had significantly higher infection frequencies than the samples collected in the summer for both DWV (X^2^ (1, N = 141) = 9.4, *p* < 0.05) and BQCV (X^2^ (1, N = 141) = 12.3, *p* < 0.05) but not for SBV.

### 3.3. Natural Survivors vs. Traditionally Managed Colonies

Figure 2 shows the percentage of virus infections (A) and the infection loads (B) for both NSCs and TMCs and for each virus. The infection frequencies were significantly higher in NSCs than in TMCs for DWV (X^2^ (1, N = 111) = 8.6, *p* < 0.05) but not for BQCV (X^2^ (1, N = 111) = 0.1, *p* = 0.75). No significant differences were found between the infection loads of TMCs and NSCs for DWV (*t* (68) = -0.6, *p* = 0.52; TMCs = 5.2 ± 0.4, NSCs = 4.9 ± 1.6) and BQCV (*t* (33) = 1.6, *p* = 0.11; TMCs = 4.4 ± 0.9, NSCs = 4.8 ± 0.1). On average, most infections hovered around 10^5^ for both DWV and BQCV in this sample cohort.

### 3.4. Queen Age

Figure 3A shows the frequency of infection with DWV or BQCV for each queen age and for both TMC and NSC groups. For DWV, a significant decrease in the percentage of infected samples was found in TMCs between queens aged 0 and 1 year (X^2^ (1, N = 59) = 3.9, *p* < 0.05). This trend continued with a lower infection frequency in queens aged 2 years (1/10) than in queens aged 1 year (14/42), albeit not significant. No differences in infection frequencies were found between queen ages in NSCs. Figure 3B shows the infection loads of DWV and BQCV for each queen age for both TMC and NSC groups. As previously shown, the infection load did not differ between the two groups. Comparing infection loads between queen ages showed significantly higher infection loads in queens aged 0 years (M = 5.7, SD = 0.2) than in queens aged 1 year for DWV (M = 4.8, SD = 1.6; *t* (29) = 2.6, *p* < 0.05) for both TMCs and NSCs. Albeit not significant, the mean DWV infection load for queens aged 2 years (M = 5.3) was lower than the mean of queens aged 1 year (M = 6.2). No significant differences in infection frequencies or infection load were found for BQCV. The infection load did however show a similar general trend, with mean infection loads decreasing with age (5.0 in queens aged 0 years, 4.6 in queens aged 1 year and 4.4 in queens aged 2 years). Interestingly, the spread in the infection loads of queens aged 0 years largely lacks infection loads lower than 10^7^ for DWV and BQCV. Seasonal differences in sample collection did not influence the infection frequency of DWV in queens aged 0 years (X^2^ (1, N = 25) = 1.69, *p* = 0.16) and older queens (X^2^ (1, N = 97) = 2.2, *p*-value = 0.14). Queen age was unknown for 19 out of 141 samples (13%).

## 4. Discussion

In comparison with the previously described SOV phenotyping protocol [49], this study implemented two improvements for virus screening in honey bee eggs. Measuring virus prevalence with qPCR showed, as expected, a higher detection sensitivity than when measuring virus prevalence with *end-point* PCR. The detection threshold of the qPCR was around 10^3^ (30 Ct) for all viruses, while the end-point PCR showed significantly higher detection thresholds for SBV, BQCV and ABPV (10^8^ for SBV, 10^7^ for BQCV and 10^6^ for ABPV). This implies that phenotyping using end-point PCR underestimated the number of virus infections for these viruses. There was no difference between both detection thresholds for DWV. It should be noted that samples negative on qPCR can still be infected below the detection threshold and that positive samples might be infected with viruses in a dormant state. An important advantage of qPCR is that breeding programs can manually set threshold values based on the breeding goal and the virulence of the virus. Each breeding program can thus determine the degree of positive or negative selection desired. Infection loads in eggs are linked with the infection status of the queen [53,54] and have been shown to reduce virus infections in the colony as a whole [49]. Nevertheless, the impact of different virus infection loads in eggs on subsequent developmental stages is currently unknown. Further research, where eggs with different virus infections are reared in vitro, could improve our understanding of the impact that vertical transmission has on antiviral responses and honey bee health.

The comparison between individual DWV genotypes and the generic DWV showed a large underestimation of DWV infections in this sample cohort when screening for either one of the genotypes or the sum of DWV-A and DWV-B. This can be seen in terms of the underestimation of the number of infected samples (25% of the samples) and the lower infection loads (on average 1.69 Log_10_ lower). In comparison, a previous study in the UK found 40% higher DWV titers when screening with a universal DWV-complex assay than when pooling the results of screening with specific DWV-A and DWV-B assays [55]. Possible explanations for this could be the presence of different genotypes, DWV-C or DWV-D, neither of which has to date been detected in Belgium [26,27], or mutations in the primer region that hamper correct primer hybridization [56]. A study on the genetic diversity within a DWV population in a colony showed that 82% of the genome had >1 sequence variant present in the frequency of >1%, and 39% of the genome had >1 sequence variant present in a frequency of >10% [8]. In addition, shifts in the sequence space of the DWV-A quasispecies have been shown after injection in honey bee pupae [57]. The rapid shifts in the DWV quasispecies are consistent with the punctuated evolution theory, whereby the infection of a new host causes a selective sweep, followed by diversification towards an increased genetic heterogeneity that has potentially adapted to the host-specific antiviral defenses [28]. This implies that primer regions, although being in conserved regions, may evolve over time and reduce the primer amplification efficiency.

With regard to the virus prevalence in drone eggs collected across Europe, virus-free samples were found in all countries and in both TMC and NSC groups in the sample cohort studied herein. According to the SOV protocol, queens laying virus-free eggs at the time of sampling were phenotyped as SOV-positive (SOV+) [49]. The low number of infections in the Gotland population (located in Näsudden) and in the TMC (located in Sigarve) populations from Sweden was remarkable, as both groups only had one sample infected with DWV despite multiple studies recording high viral loads in the worker bees of Gotland throughout the years [35,38,44,45,58,59,60]. The presence of SOV+ queens across Europe serves as a possible starting point for local breeding programs to perform selection within the variation in virus resistance present within honey bee populations [49,61]. Including subsequent generations descending from SOV+ queens in breeding programs is crucial to maximize selection on the heritable genetic contribution [49] behind virus resistance. This is because the SOV trait does not differentiate between samples that are free of viruses due to the virus resistance of the queen or due to other non-genetic circumstances. In this study, virus abundance and prevalence were based on one sampling time point. However, for accurate SOV phenotyping, multiple samples could be collected at different time points to differentiate between queens that temporarily lay virus-free eggs and queens that do so consistently. This is an important consideration, as virus dynamics have been shown to change across the seasons [62].

In addition to the high prevalence of DWV, the second most common virus found in this study was BQCV. This virus is the most common cause of queen larval death [63,64], but it has not been found to cause overt symptoms in queens despite the detection of high infection loads [65]. Viruses of the ABPV complex and SBV have been found in eggs [54,66,67,68], but they were rarely detected in this study. The higher virulence of BQCV, SBV and ABPV [4,48,69,70,71,72] compared to the lower virulence of DWV [69] could explain why they are less likely to be transmitted vertically without causing queen supersedure or colony health issues [73].

Differences in infection patterns between countries can be caused by climatic conditions, by the seasonality of honey bee viruses [34,56,74,75,76,77,78,79,80] or by the low number of samples per country. This is reflected in the significant differences in infection frequencies between the spring and summer sampling seasons in this study. Interestingly, infection frequencies with DWV were higher in spring, contrasting with the generally higher infection frequencies reported for adult bees during the summer and autumn [62,75,77,80]. These findings indicate that conclusions based on SOV phenotyping should always take the time of sampling into account when interpreting results and that the SOV status can change between seasons. Other factors affecting virus abundance are nutritional quality and availability [81,82], connectivity between colonies [83], colony demography [84,85], population heterogeneity [86,87,88,89,90], colony management [91], the degree of local adaptation [92], individual and colony-level immune responses [93] and other stress factors (such as exposure to neonicotinoids) [94]. In this study, worker eggs did not have higher infection loads than drone eggs despite the possible occurrence of trans-spermal virus transmission [25].

By comparing naturally surviving with traditionally managed populations in the same local context, insights can be gained into which evolutionary adaptations are needed for honey bees to survive without treatment against the Varroa mite. Typical for naturally surviving populations is that they harbor higher mite numbers that serve as an important vector for viruses [39,95]. Honey bee colonies react to these high disease pressures with adaptations in their antiviral responses or by forms of social immunity [96]. This study shows that, in this sample cohort, infection frequencies were significantly higher in NSCs than in TMCs for DWV but that infection loads did not differ between the two groups. Honey bee queens appear to avoid increased vertical transmission loads despite increased infection frequencies. Virus loads in worker bees were not studied here. Therefore, it remains uncertain if the higher infection frequencies are a result of increased virus circulation in the naturally surviving populations. The high variability between both groups in each country could be caused by previously mentioned factors affecting virus abundance, the time since colonies were left untreated [36,37] or the degree of genetic divergence between TMCs and NSCs within a country [52,62,90].

Honey bee queens accumulate viral infections and infection loads during queen rearing [97,98], during mating flights [47,99,100] or as they become older [47,51,68], despite increased immune responses [101]. In contrast to what was expected, both the infection frequency of DWV and the infection load in the eggs of queens from TMCs decreased with increasing queen age. This difference is not caused by the mortality of queens with high infection loads, as the distribution of the egg infection loads does not overlap between queens aged 0 years and 1 year. Queens from NSCs showed the same trends in infection loads across all queen ages but did not show differences in infection frequencies. For BQCV, only the infection load was significantly lower in queens aged 0 years than in queens aged 1 year from TMCs. In beekeeping practices, queens are often renewed yearly, as young queens are associated with lower winter mortality [66,102]. This study suggests that older queens from colonies that are treated against the Varroa mite might be able to adapt their antiviral responses to DWV and thereby reduce the infection loads transmitted via their eggs. If so, the frequent renewal of queens could limit this potential as opposed to selecting towards increased queen longevity.

By focusing on the role of honey bee queens, this research adds to the growing literature on the relationship between viral infections and honey bee health. Evolutionary patterns of resistance and tolerance can form the theoretical foundation to incorporate virus resilience in breeding programs. This is a promising perspective, as shown by the variability of vertical transmission over time, across queen ages and under different evolutionary conditions.

## Figures and Tables

**Figure 1 viruses-14-02442-f001:**
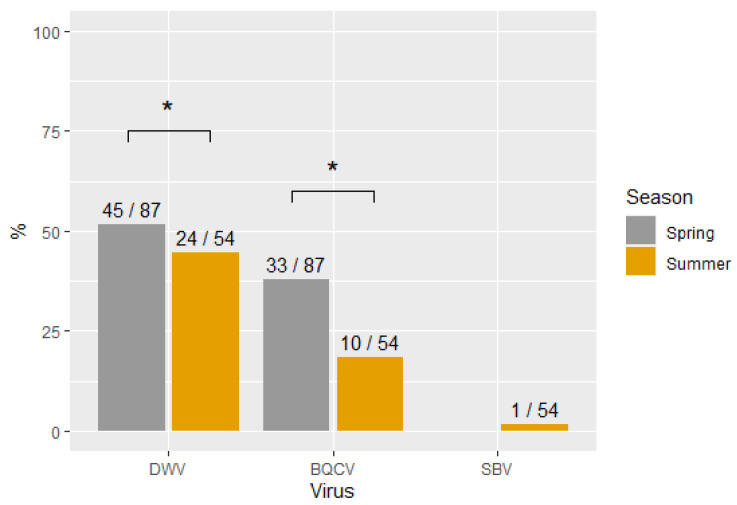
Percentage of virus infections in each sampling season. Significant differences are indicated with *. The numbers on top of each bar represent the number of positive samples and the total sample for each group.

**Figure 2 viruses-14-02442-f002:**
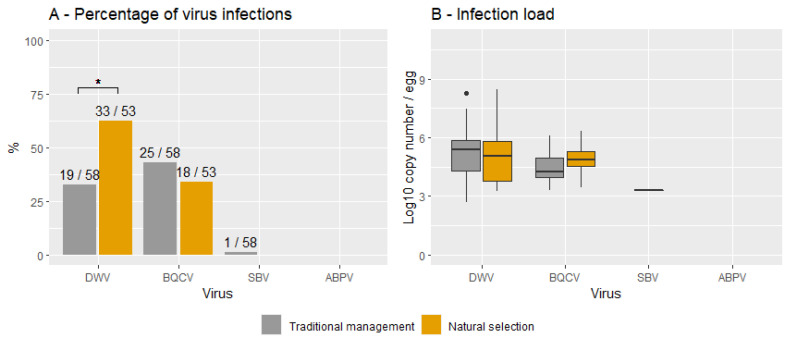
Percentage of viral infections (**A**) and infection loads (**B**) for naturally selected and traditionally managed colonies. Data are provided for each virus. Significant differences are indicated with *. The numbers on top of each bar represent the number of positive samples and the total sample for each group.

**Figure 3 viruses-14-02442-f003:**
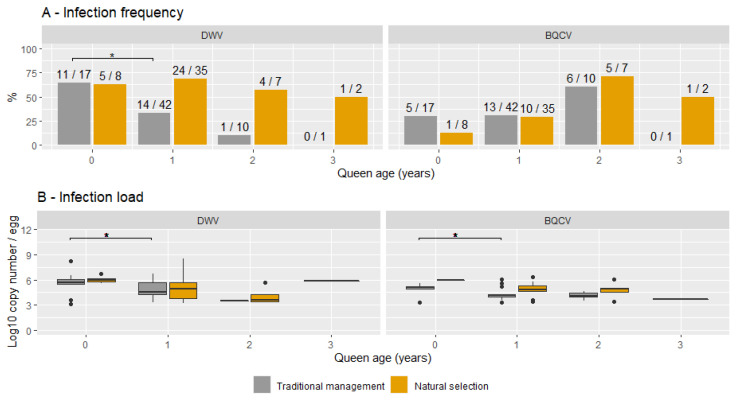
Overview of the infection frequency (**A**) and the infection load (**B**) for each queen age from both naturally selected and traditionally managed colonies. Significant differences are indicated with *. The numbers on top of each bar represent the number of positive samples and the total sample for each group.

**Table 1 viruses-14-02442-t001:** Virus prevalence in pooled egg samples from each participating country for both naturally surviving colonies (NSCs) and traditionally managed colonies (TMCs).

Country	Sampling Season	NSC/TMC	No. of Sampled Colonies	No. of Virus-Free Samples	No. of Samples Positive For:				Mean Infection Load (Log10/Egg)			
					DWV	BQCV	SBV	ABPV	DWV	BQCV	SBV	ABPV
Belgium	Spring	NSC	10	1	9 (90%)	0	0	0	4.3			
		TMC	11	4	7 (64%)	2 (18%)	0	0	6.1	4.2		
Croatia	Summer	TMC	10	3	4 (40%)	3 (30%)	1 (10%)	0	4.7	5.3	3.3	
France	Spring	NSC	13	2	7 (54%)	11 (85%)	0	0	5.6	5.5		
		TMC	10	2	1 (10%)	8 (80%)	0	0	5.8	6.4		
the Netherlands	Partly in spring and summer	NSC	10	2	8 (80%)	3 (30%)	0	0	6.2	5.4		
		TMC	6	1	5 (83%)	3 (50%)	0	0	5.4	4.9		
Norway	Summer	NSC	10	1	9 (90%)	1 (10%)	0	0	5.1	4.9		
		TMC	10	5	4 (40%)	3 (30%)	0	0	4.3	4.7		
Portugal	Spring	TMC	10	1	8 (80%)	1 (10%)	0	0	5.1	3.3		
Romania	Spring	NSC	4	1	0	3 (75%)	0	0		4.5		
		TMC	9	4	2 (22%)	4 (44%)	0	0	4.0	4.0		
Slovenia	Spring	TMC	72	27	38 (53%)	11 (15%)	2 (2%)	2 (2%)	5.3	5.2	3.2	3.8
Spain	Spring	TMC	10	4	5 (50%)	1 (10%)	0	0	5.5	4.8		
Sweden	Summer	NSC	6	5	1 (16%)	0	0	0	5.0			
		TMC	12	11	1 (8%)	0	0	0	4.3			

## Data Availability

The data presented in this study is openly available in FigShare at https://doi.org/10.6084/m9.figshare.21485865.

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
