# Peer review of "Virus Prevalence in Egg Samples Collected from Naturally Selected and Traditionally Managed Honey Bee Colonies across Europe"

_viruses, 2022, doi:10.3390/v14112442_

Round 1

Reviewer 1 Report (Previous Reviewer 2)

The manuscript has improved substantially and the comments and suggestions made in the previous version of the manuscript have been taken into account, so I believe it can be accepted for publication.

Author Response

No comments or suggestions were requested. 

Reviewer 2 Report (New Reviewer)

1.The general situation of all viruses tested should be indicated in the abstract, such as SBV and ABPV, though the infection frequency is very low.

2.In line 157, 101-108/10 eggs, should be subsituted by 101-108/10 eggs.

3. In lin 241,reference [68] should be numbered [53].

Author Response

Dear reviewers, 

First of all many thanks for your valuable comments. 
Please find the responses to the comments below. 

1.The general situation of all viruses tested should be indicated in the abstract, such as SBV and ABPV, though the infection frequency is very low.
Added the following sentence in Line 45: ‘Of the 213 screened samples, 109 were infected with DWV, 54 with Black queen cell virus (BQCV), 3 with Sacbrood virus and 2 with Acute bee paralyses virus.’

2.In line 157, 101-108/10 eggs, should be substituted by 101-108/10 eggs.
Adjusted. 

3. In line 241, reference [68] should be numbered [53].
Adjusted.

This manuscript is a resubmission of an earlier submission. The following is a list of the peer review reports and author responses from that submission.

Round 1

Reviewer 1 Report

Editors,

The manuscript entitled, “Dynamics in vertical transmission of viruses in naturally selected and traditionally managed honey bee colonies across Europe” by Bouuaert et. al describes virus presence and abundance information from honey bee egg/embryo samples taken from ~ 10 colonies at two different dates within a single season/year (i.e., spring and summer) from 9 different countries, and therefore represents about 200 samples – although the description of the sample cohort in the methods, text, and supplemental table should be made clearer and include more details. The data is interesting since there have been relatively few studies that have quantified virus abundance in honey bee embryos, and it provides some basic initial data. The authors obtained samples from “traditionally managed (i.e., treated with miticides) (TMC) colonies” and “non-miticide treated colonies” which they term naturally surviving colonies (NSC), but the authors provide no information on the details of these colonies (i.e., for how many years have they survived, etc.). The introduction sets up that this study will provide information on honey bee host, and virus evolution, but the data provided in this paper is insufficient to assess evolutionary trends. Perhaps a better description of the sample cohort and data presented at the colony level (i.e., colony x, location XXX, spring sample results, summer sample results, colony strength, colony age/survival, etc.) would be more informative to investigating the ideas/theories introduced in the introduction, but as written this is not the case – and furthermore much of the data is variable and thus not statistically significant. That said, the data is a stepping stone to important studies.

Points to clarify or address before publication include:

1. Abstract – Line 34 – The statement “virus-free status” of eggs associated with virus-resistance at the colony level” – if this statement represents data from a single study- it seems that is an overstatement given the available data (i.e., how many times was this found? what viruses? etc.

2. The authors should make it clear – early in the manuscript -  how many times were colonies monitored and the dates at which samples were obtained (i.e., for each colony).

Virus abundance at the colony level has been strongly associated with date

for example see: Faurot-Daniels et al PONE 2020

https://doi.org/10.1371/journal.pone.0237544

and in general colony-level virus data varies by date – so it is very important to include exact dates and locations for each sample collected in these types of studies.

3. Table 1. It seems based on the variability in this data set that the total number of naturally selected colony or traditionally managed colony samples are too few. The authors should make it very clear how many of each were included in the study. They could also provide advice (i.e., based on their experience) how many colonies should be monitored for future studies (i.e., maybe 50 of each management style at two different time points within the same geographic location and on the same dates?).

What is the next best step?  Or perhaps a smaller number of colonies should be monitored for years, and samples obtained from queen daughters and their colonies etc. – to begin to assess selection pressures.

4. Table 1 and Figure 3 B – log(1) per egg should also include per ng RNA, since qPCR measures SQ of cDNA, and it may be that not all samples had similar levels of RNA extraction (i.e., due sample variation) or include RNA data and/or data from housekeeping gene.  This information is important to be able to compare egg/embryo virus levels to levels in adult bee samples etc.. Articles in Virology journals, typically include qPCR data as “copies per XXX ng RNA”.

5. Figure 1.  Data is interesting since the embryo data shown is different than the trends observed from studies of adult bees (i.e., US, Canadian, Italian, and German studies that show DWV levels typically increasing from spring to summer, and reaching maximum levels in fall – see Traynor, et al. 2017 Apidologie https://doi.org/10.1007/s13592-016-0431-0; Bee Informed Partnership, Canadian National Honey Bee Health Survey (see: https://profils-profiles.science.gc.ca/en/publication/canadian-national-honey-bee-health-survey-2014-2017), Faurot-Daniels et al. (. doi: 10.1371/journal.pone.0237544), Glenny et al.( https://doi.org/10.1371/journal.pone.0182814, Genersch et al. Apidolgie DOI: 10.1051/apido/2010014. This difference would be worth discussing further in this manuscript.  

6. Lines 13-114 – It is good that the authors clearly define the ‘suppressed in-ovo virus infection trait” since it is not something that virologists would generally be familiar with, but this section should also clearly state that “drone eggs” are unfertilized (and therefore reflect the vertical virus transmission by the queen bee only), and maybe clarify that drones are male bees. It may also be important to state that drone brood is only present in the spring and summer (and not the fall), therefore an additional sample could not be obtained (if this is correct).

7. Since the data presented in this manuscript represents data from two points in time in 10 colonies per participating country, the abstract, introduction, and text should reflect that the data and findings represent “this particular sample cohort”, but until additional studies that support this data are carried out – it shouldn’t be presented as overall, generally true findings.  The authors are encouraged to add “in this sample cohort” at least a few places in the text to indicate the limitations of their conclusions (i.e., they are from this snapshot of data).

8. Supplementary Data

Supp. S2. The virus detection range of 10^1 – 10^ 8 per egg, should be changed to amount of RNA (since RNA extraction per egg may vary) or at least include an average RNA per egg value, which would allow comparison with other research studies. The quality of the gels shown in this figure could also be improved.

9. The linear equations for qPCR quantification for each virus should be included see

for example see: Daughenbaugh, et al Viruses 2021 Supp. Data for example: DOI: 10.3390/v13020291)

“Plasmid standards, containing 103 to 109 copies per reaction, were used as qPCR templates to assess primer efficiency and generate standard curves used for quantifying viral RNA copy numbers for each sample. Primer efficiencies were evaluated using qPCR assays of cDNA and plasmid dilution series, and calculated by plotting log10 of the concentration versus the crossing point threshold (C(t)) values and using the primer efficiency equation, (10(1/Slope)-1) × 100) [109]. The virus-specific qPCR primer sets utilized in this study had efficiencies ranging from 86 to 103% and provided accurate quantitative assessment of >103 RNA copies per reaction. The linear standard equations for virus-specific plasmid standards and qPCR primers listed in Supplementary Table S3 were as follows: Rpl8 Ct = -3.937× + 40.506, R2 = 0.9982; LSV Ct = -3.252× + 39.697, R2 = 0.9979; DWV Ct = -3.4093× + 41.641, R2 = 0.99756; AmFv Ct = -3.3185× + 39.489, R2 = 0.99151; BQCV Ct = -3.2918× + 40.332, R2 = 0.99378; SBV Ct = -3.725× + 44.017, R2 = 0.99018; AnBV-1 RNA 1 Ct = -3.5923× + 42.246, R2 = 0.99893; AnBV-1 RNA2 Ct = -3.3896× + 44.259, R2 = 0.94813; negative strand AnBV-1 RNA1 Ct = -3.417× + 40.427, R2 = 0.9995; negative strand AnBV-1 RNA2 Ct = -3.5973× + 44.846, R2 = 0.9972. All reported qPCR data were normalized to genome copies per 100 ng RNA.

10. Abstract -  Line 45-46  “Together, these patterns in vertical transmission suggest an adaptive antiviral response of queens aimed at reducing vertical transmission over time.” should be revised since this is an overstatement  - this is not shown by the data in this paper, which does not include in immune gene expression data, in queen bees, embryos, or next generation of bees.

11. Introduction Lines 52-57 should be revised. In general honey bee viruses were not studied as much before the arrival of Varroa destructor and based on the literature, it is not accurate to report such a general statement that “virus infections were benign before mite introduction” and provide one citation from 2004 with mathematical modeling data. This statement should be revised or removed. There is still a lot to learn about the impacts of viruses on honey bees (particularly at the colony and individual bee levels) and much of this research will depend on quantitative data (i.e., as opposed to the assessment of presence and absence of a few viruses associated with a colony level sample in samples collected at one point in time).  

12.  Introduction Lines 76-84+

It would be good to provide an indication of how long (i.e., how many years) the Gotland honey bee colonies have been studied/monitored – since the authors are describing or indicating “honey bee and or virus evolution” it is important that the readers have a sense of the “evolutionary time period” referred to in this paragraph. Please include relative average colony / queen lifespans and indicate other examples of selection in insects or other organisms within similar time frames. etc..  Also discuss, include information regarding “colony-level” selection in honey bee colonies highlighting that the queen is the sole reproductive unit, role of drones, etc..  This general discussion glosses over many important details that are important for choosing to frame this paper//introduction using an evolutionary framework – especially given that this study does not include samples from Gotland colonies, and only presents data from a single bee season. The authors should think more about the appropriateness of extensive coverage of these colonies / this topic given the data presented in this manuscript. Do the Natsudden samples in this study include / represent Gotland honey bee colonies?

13. Clarify this sentence: Line 93  “In a honey bee colony, resistant individuals will lead to a resistant colony.” It seems more appropriate that resistant individuals will ensure a colony survives – but in the context of selection and the rest of this paragraph, it is important to the reproductive member of the colony is resistant. 

14. Supplemental Table and manuscript text should better indicate if the Natsudden samples are from the Gotland colonies that were extensively described in the Introduction.

15. Lines 98-100 As written, it is confusing how “high Nosema infection loads would result in clearance of infection.  Nosema levels in colonies in the US/Canada are prevalent and abundant (see Bee Informed Partnership and Canadian Honey Bee Monitoring Project Data) and other studies - … although this data isn’t from a breeding program, the high levels even in the absence of treatment do not suggest clearance is the next step. Although, without data it is difficult to speculate. This section cites ref 53, which is a review on honey bee queens, without reading that paper too, it is not clear what data support Nosema clearance therefore it may be important to also include the primary citations for these statements.

16. Lines 120 – 121 + include length of time of selection for all NSC and TMC colonies; this could maybe be added to Table 1 or Supp. Table 2.

17. Line 130 – consider revising “evolutionary setting” to “breeding program selection strategies” since as described above the evolutionary time frame for the honey bee host is not clear in the context of this study and others.

18. Methods – SOV phenotyping. More details are needed on the NSC, colonies that survive without Varroa treatment. They are insufficiently described for readers of this manuscript to evaluate the “Darwinian Black Box” or Bond Test selection – and those should also be described again in the methods too, since these are not common knowledge tests.

19. Line 156 – The colony health data obtained at each sampling event (i.e., frame count, mite count data, etc.? ) should be reported as part of supplementary information.

20. To address a few points, it may be best to add a section in the results that is a more thorough and accurate description of the sample cohort and associate that description with a supplemental table with the additional information requested in this review. This would also facilitate linking the some of the material in the introduction, with the samples obtained for this study (if applicable). Were samples obtained from the same colonies in spring and summer or just from the same apiaries?  Data should also be presented for each sample for each colony (i.e., in a an excel sheet) since 50% of the samples showing opposite trends would look like “no change”, but it would be informative to know if the infection status differed and by how much and what direction in samples obtained from a single colony at two time points.

21. Methods – add “Reverse Transcription or cDNA synthesis” and remove that step from PCR and qPCR, or describe differences in the RT if there were differences.

22. Methods 2.3 “Gel-based” RT PCR should be “PCR (or End-Point PCR)” since reverse transcription was carried out using random-hex primers (i.e., not virus or gene-specfic primers) in a separate reaction prior to PCR.

Typically, “RT-PCR’ is used when the steps are carried out in one reaction (ether one- or two step RT-PCR). This should be corrected throughout manuscript.

23. Methods – 2.4 should just be “Quantitative PCR (qPCR)” unless the RT step was carried out in the same wells as the qPCR reaction Promega resource is useful: https://www.promega.com/resources/student-resource-center/lab-essentials/nucleic-acid-amplification/

24. Discussion:  Revise ~ Lines 325+

It is also possible that at a given “snapshot” in time a queen is “virus-free” due to temporary clearance of virus infections. This possibility should also be included rather than only the evolutionary theoretical discussions described. The authors do not present enough data in this paper to state that a “virus free drone egg samples” at one or two points in time in the long life of a queen bee indicates a virus-resistant queen. While this paragraph makes this point in Lines 333-334 the text before it is misleading and an overinterpretation of the data obtained.

25. Line 344- Colony level studies across the globe have associated high DWV loads and high mite infestation as correlates to poor colony health and/or overwinter colony deaths – so it seems inaccurate to described DWV as having “low virulence” .

26. Line 348 – and likely early should cite Faurot-Daniels et al, which indicates that sample collection data is strongly associated with virus presence and abundance.

27. Lines 382+ discuss a non-significant results “a trend in BQCV”  in NSC queens, therefore This study suggests that older queens are able to adapt their antiviral responses and reduce the infection loads transmitted via their eggs. If so, frequent renewal of queens could limit this potential as opposed to selecting towards increased queen longevity. “Therefore, although interesting and potentially correct – it can not be put forth here since the data in this study (Fig. 3) do not support this conclusion.

28. Discussion Lines 397+ The data, which do not include any assessment of queen immune gene expression, nor embryo gene expression are insufficient to suggest immune priming – this section should be deleted or revised. As written, it is confusing because it seems the authors are claiming that their data are in-line with this theory, but the data presented in this paper is not statistically significant.  

29. Lines 410-411 – seem outside the scope the data shown.

Minor points to clarify or address before publication include:

1. Italicize “Varroa destructor” throughout manuscript.

2. Abstract: Line 38 included full name of virus (i.e., deformed wing virus (DWV)).

3. Figure 3 – horizontal axis should include (years)”; so of the spacing in the chart is off (i.e., in the BQCV 13/42, etc.)

4. Supp. Fig. 3 – include exact sampling dates in this table, make sure all locations are towns and not regions.

Include additional details, as pointed out in other parts of this review. An excel document may be best.

Since the colony health descriptions should be associated with each sample, and not just listed and “1 out of 10 colonies had foulbrood” etc.

5. Abstract Lies 43-45 Suggest changing the following sentence to enhance clarity:

 “Seasonal variation in vertical transmission was found with lower infection frequencies in spring compared to summer for DWV and black queen cell virus.”

Suggested revision: “We determined that the detection frequency of honey bee egg associated DWV and BQCV was lower in samples obtained in the spring compared to those collected in the summer, indicating that vertical transmission may be lower in the spring.”

Of course, there are many alternative revisions that would help clarify the intended meaning of this sentence.

I found a few other sentences difficult to read / unclear.

For example,

a. Introduction Lines  50-51

b. Lines 61-62, although the intended meaning is sequence variants of either ABPV or DWV virus, as written – it could be interpreted incorrectly that these two viruses recombine; suggest revising.

6. Lines 206-207 fix superscripts, or correct text since it is unlikely that the detection limits are so precise (i.e., 106 vs. 107 copies)?  Or include complete linear range for all viruses with equation.  Maybe lines 290-291 too.

7. Line 254 “5 Log 10” should be corrected, that is not normally how numbers are written, 105 would be better (or

at least use subscript e.g., (5 log10), but this is odd.

8. Throughout “X2” should be “ χ2  

9. Line 310 – fix “bibliography”

10. Line 353 “time” should likely be “date” or “day within typical bee season”

Reviewer 2 Report

The article under appreciation is an original contribution relevant for the Viruses journal. The study focuses on determining the dynamics in the vertical transmission of some of the main viruses that affect bees across Europe. The study compares the presence of viruses in eggs collected from traditional manage colonies and colonies that survived naturally. Likewise, the results considered the influence of the queen age and the season sampling over the infection in the colony eggs.

The study is interesting, but the manuscript needs further elaboration and considerable modifications.

Please consider the following aspects:

1 Integrate a separated section for RNA extraction and cDNA synthesis in materials and methods.

e.g.     

  2.3 RNA extraction and cDNA synthesis

                               2.4 Gel-based RT-PCR

                               2.5 RT-qPCR

                              2.6 Statistics

2 In the results section (lines 205-207) detection threshold of the RT-PCR test for each virus are reported. These results are supported by images of gels in the supplementary material section S2. However, the gels images for each virus looked overlapping from 2 or more gel images. For greater clarity in the results, I recommend presenting a single image per virus, which includes the DNA Ladder, the positive and negative controls, and the amplifications with the detection threshold.

3 Table 1: In some cases, the total number of samples analyzed does not correspond to the number of virus-free and positive samples. Therefore, the percentages of positive samples may be wrong.

e.g.

Nr. of samples positive for:

Nr. of samples positive for:

Country

NSC /

TMC

Nr. of samples

Nr of virus-free

samples

DWV

BQCV

SBV

ABPV

France

NSC

13

2

7 (58%)

11 (85%)

0

0

TMC

10

2

1 (10%)

8 (80%)

0

0

4 Results related to figure 2 are described. However, figure 2 is missing from the manuscript.

5 Minor comments:

Line 38: please substitute “DWV genotypes” with “deformed wing virus (DWV) genotypes”

Lines 44-45: Please include the abbreviation “black queen cell virus (BQCV)

Line 52: please substitute “Apis mellifera” with “Apis mellifera

Line 52: please replace “Varroa destructor” with “Varroa destructor

Lines 60-61: remove italics from virus names

Lines 139-140: please substitute “101-108 / 10 eggs” by “101-108 virus / 10 eggs”

Lines 243-245: Remove from figure caption: “Infections were significantly more frequent in samples collected in spring for DWV and BQCV as compared to samples collected in summer.” The information is already in the text.

Line 310: remove “{Bibliography}”

Line 400: please substitute “[reviewed by 104,106,108]” by “(104,106,108)

Lines 716-720: references 121 and 122 are missing in the manuscript.